# OVERFITTING DETECTION OF DEEP NEURAL NETWORKS WITHOUT A HOLD OUT SET

## ABSTRACT

Overfitting is an ubiquitous problem in neural network training and usually mitigated using a holdout data set. Here we challenge this rationale and investigate criteria for overfitting without using a holdout data set. Specifically, we train a model for a fixed number of epochs multiple times with varying fractions of randomized labels and for a range of regularization strengths. A properly trained model should not be able to attain an accuracy greater than the fraction of properly labeled data points. Otherwise the model overfits. We introduce two criteria for detecting overfitting and one to detect underfitting. We analyze early stopping, the regularization factor, and network depth. In safety critical applications we are interested in models and parameter settings which perform well and are not likely to overfit. The methods of this paper allow characterizing and identifying such models.

## 1 INTRODUCTION

Deep neural networks have shown superior performance for a wide range of machine learning task such as speech recognition (Graves & Jaitly (2014)), image classification (Krizhevsky et al. (2012)), playing board games (Silver et al. (2016)); machine translation (Kalchbrenner et al. (2016)); beating previous methods by orders of magnitudes. To apply neural networks to safety critical problems such as autonomous driving it is necessary to evaluate their performance on new previously unseen data.

One of the major problems of neural networks is their vulnerability to adversarial attacks. It has been shown that tiny unrecognizable changes of the input can fool the network to predict any class the attacker has chosen. One way to interpret this vulnerability is that the neural network overfits to the training data, with the output varying rapidly around each training point and thus slight changes of the input can lead to big changes in the output. It is thus highly desirable to prevent the network from overfitting during training.

Previously reported methods reduce the chance of overfitting by evaluating the neural network on some holdout set, or by penalizing the complexity of the model class. This has the disadvantage that a holdout set can only be used once. By using design choices proven to be successful in the past the model becomes dependent on the holdout set. Penalizing the model class is only a heuristic remedy to overfitting. In the present paper we devise a method which prevents overfitting by relying on the training data only. We motivate $l_1$-regularization of the kernel weights as a preferable choice to control the network complexity. Using no holdout set requires an alternative notion of overfitting. In the paper, we say that a model overfits if it is able to learn noise.

## 2 LITERATURE REVIEW

**Heuristics.** There exists several well known heuristics which reduce the chance of overfitting. Typically one reduces the hypothesis space, or one increases the data set. We can directly control the size of the hypothesis space by the number of parameters of the model. Typical choices are the *width*, the *depth* and the *filter size* of the network. *Dropout* introduced by Srivastava et al. (2014) at each training step, individual nodes and their incoming and outgoing edges are ignored (dropped out) with probability $p$. This reduces the dependency of the model on individual nodes. In *early*

*stopping* (Prechelt (1996)) the model complexity is controlled by the number of iterations, in *weight decay* (Krogh & Hertz (1991)) an $l_2$ penalty on the weights is added to the loss function.

**Data augmentation.** *Data augmentation* is a commonly used technique to prevent overfitting. Here the size of the input data is artificially enlarged by applying a *transformation* to the input signal which is supposed to keep the output fixed. In practice this is done by *cropping*, *adding input noise*, applying *affine transformations* or *small deformation* to the input and so on. A more indirect data augmentation technique is *input normalization* see for example Jaderberg et al. (2015).

**Detecting overfitting.** There are several known approaches to detect overfitting. In *holdout testing* the data set is split in *train* and *test* sets. The models are learned on the training data only. After training the models performance is tested with the test data, thereby providing an empirical estimate of the true risk. If data is scarce *cross validation* techniques can be applied.

**Generalization theory.** Generalization theory relates the minimization of the empirical risk to the minimization of the (unknown) expected risk. The difference between both risks is called the generalization gap, see Kawaguchi et al. (2017). An important result by Bartlett (1998) proves that the $l_1$-norm of the weight matrices is more important for generalization than the number of weights. Furthermore, in Bartlett et al. (2017) it was shown that the spectral complexity of the neural network bounds the generalization gap. This norm based bound is independent of the architecture parameters, except for log terms.

**Overfitting and neural networks.** For neural network not all of these techniques behave as one would expect. For example it has been reported that deeper networks generalize better than shallower networks. It seems that the number of parameters of a network is of lesser importance to generalization as measured on the test set. Several large scale experiments supporting this claim are reported in Figure 1 of Novak et al. (2018).

**Randomization experiments.** Several papers used randomized labels to investigate generalization and memorization in neural networks. Most notably Zhang et al. (2017) showed that a neural network can memorize randomized training data well, concluding that a theory based on the complexity of the model space alone can not explain the generalization puzzle. Arpit et al. (2017) analyzed memorization in deep networks based on randomization of the training data.

## 3 METHODS

### 3.1 COMPLEXITY AND $l_1$-REGULARIZATION OF THE KERNEL WEIGHTS

The complexity of a neural network is controlled by its hyper parameter and the hyper parameter of the training algorithm. We propose to control the model class by adding the $l_1$ norm of the kernel weights $||W_j||_1$ multiplied with a regularization factor $\lambda$ to the loss function. Some additional background for this section is provided in the appendix.

**Notation.** This paper considers feed forward networks $f: \mathbf{R}^{d_0} \to \mathbf{R}^k$ which maps the *input space* $V_0 = \mathbf{R}^{d_0}$ to some *target space* $V_L = \mathbf{R}^k$. This is followed by an $\arg\max$ function which picks as output the coordinate of the largest value. The *margin* measures the gap between the output for the correct label and the other labels, $\gamma = f(x)_y - \max_{j \neq y} f(x)_j$. A positive margin means a correct classification, whereas a negative margin means an incorrect classification.

Each layer of the network consists of a linear block $A_k: V_{k-1} \to W_k$ and a non linearity $\phi_k: W_k \to V_k$. The networks are thus written as:

$$f(x) = \phi_L \circ A_L(\cdots(\phi_1 \circ A_1(x)))$$

Here of course $\phi \circ A(x)$ is just another way of writing $\phi(A(x))$. The concatenation $\eta_k = \phi_k \circ A_k$ will be called a *block* of the network. In this paper only the standard *relu* nonlinearity $\phi: \mathbf{R} \to \mathbf{R}^+$, which is defined by $\phi(x) = x \vee 0 = \max\{x, 0\} = x^+$, is considered. The *input data* is denoted by $x_0 \in \mathbf{R}^{d_0}$. Further, the *output of layer* $k$ of the network is denoted by $y^k = f_k(x) := \phi_k \circ A_k(\cdots(\phi_1 \circ A_1(x))) \in \mathbf{R}^{d_k}$ and the network which outputs the $k$-th layer by $f_k: \mathbf{R}^{d_0} \to \mathbf{R}^{d_k}$.

Finally, the *width* of the network is defined by $d = \max\{d_0, \ldots, d_L\}$. In the paper we will call the data $y_k$, passing through the layers, *signal*. Finally, we arrange all data points and signals in matrices, denoted by $X \in \mathbf{R}^{d_0 \times n}$ and $Y^k \in \mathbf{R}^{d_k \times n}$. So we write by slightly abusing notation, $Y^k = f_k(X) = f_k(x_1, \ldots, x_n) = [f_k(x_1), \ldots, f_k(x_n)]$.

$l_1$-**regularization**    A typical convolutional kernel $W$ is determined by the filter size, and the number of incoming and outgoing features. If the convolution is written as matrix operation $y = Ax$ and *zero padding* is being assumed, then the matrix $A$ can be arranged as a vertically stacked block matrix each subblock $A_i$ representing one outgoing feature. The entries of each these blocks are determined by the weights of the $i$-th filter. Due to zero padding and weight sharing, each weight occurs precisely once in each row and each column of $A_i$. It follows that the filter matrix $A$ contains in each column each weight precisely once.

**Lemma 3.1.1.** *The spectral norm of a convolution matrix $A$ is bounded by the $l_1$-norm of its kernel weights.*

$$||A||_\sigma \le ||W||_1$$

*Proof.* The inequality follows as the spectral norm can be bounded by the row and columns norms of $A$, which can be estimated by the weight matrix $W$.

$$||A||_\sigma \le \sqrt{||A||_{1\to1}||A||_{\infty\to\infty}} = \sqrt{\max_{j\in\{1,\ldots,n\}} \sum_{i=1}^{m} |a_{ij}| \max_{i\in\{1,\ldots,m\}} \sum_{i=1}^{n} |a_{ij}|} \le ||W||_1 \qquad (1)$$

$\square$

Recently shown generalization bounds are dominated by $\frac{R_A ||X||_2}{n\gamma}$ (Bartlett et al. (2017)) see also Theorem A.1.1. Here $\gamma > 0$ is the *margin*, $X$ denotes the training data arranged in a matrix, and $R_A$ is the *spectral complexity*. We will use a simplified version of $R_A$ defined by

$$R_A := \prod_{j=1}^{L} ||A_j||_\sigma \left( \sum_{i=1}^{L} \frac{||A_i^T||_1^{2/3}}{||A_i||_\sigma^{2/3}} \right)^{3/2}. \qquad (2)$$

**Lemma 3.1.2.** *The spectral complexity can be bounded by the $l_1$-norm of the kernel weights $W_j$.*

$$R_A \le dL^{3/2} \prod_{i}^{L} ||W_i||_1$$

*Proof.*

$$R_A = \left( \sum_{i=1}^{L} \prod_{j\neq i}^{L} ||A_j||_\sigma^{2/3} ||A_i||_1^{2/3} \right)^{3/2} \le \frac{L^{3/2}}{L} \sum_{i=1}^{L} \prod_{j\neq i}^{L} ||A_j||_\sigma ||A_i||_1 \le d\sqrt{L} \sum_{i=1}^{L} \prod_{i}^{L} ||W_i||_1 \quad (3)$$

$$\le dL^{3/2} \prod_{i}^{L} ||W_i||_1 \qquad (4)$$

Here the first inequality holds because of inequalities between the generalized 2/3-mean and 1-mean. The second inequality follows as the spectral norm can be bounded by $||W||_1$ by Lemma 3.1.1. $\square$

Both lemmas show that it is beneficial to $l_1$-norm of the kernel weights. In fact Lemma 3.1 shows by looking at $\frac{R_A ||X||_2}{n\gamma}$ that the margin scales as $||W||_1$. Since we do not use a bias in our model, we may rescale each kernel matrix by $||W||_1^{-1}$. This is compensated by decreasing the margin accordingly. So in order to achieve better generalization bounds we can penalize the kernel weights by $\lambda ||W||_1$. Here $\lambda > 0$ is a regularization parameter to be determined.

## 3.2 ACCURACY CURVES

**Assumptions.** To simplify the analysis we make three *assumptions*. First, we assume that the data is independent and identically distributed. This implies that the with an increasing level of randomness the complexity of the data also increases. In dependent data, this is not necessarily the case. As in that case correlation in the data can be destroyed by the introduction of randomness making the data faster to learn. Second, we assume that the model complexity is controlled by a *regularization parameter* in such a way that an increase of the regularization parameter implies a strict decrease of the complexity of the model. Third, we assume that the regularization parameter and the randomness are on a similar scale. To explain this, note that the accuracy is a function of the regularization parameter and randomness. The assumption simply means that the norm of the gradient of the accuracy taking in direction of the regularization is of the same order as the norm of the gradient taken in direction of randomness.

**Creating randomized data.** In this paper we consider a classification problem. Each data point consists of a pair $(x, y)$ which is typically an image and a corresponding class. For a fixed level of randomness $p \in [0, 1]$ we split the training data $Z = (X, Y)$ in two parts. The first split $Z_p$ contains a random sample of $p$-percent of each class of the training data. The second part $Z_{1-p}$ contains the rest of the training data. The classes of the first set are randomly permuted to give the set $\tilde{Z}_p$. The randomized data is obtained by joining $D_p = \tilde{Z}_p \cup Z_{p-1}$. With this $D_0$ is equal to the original data set $Z$ and $D_1$ is obtained by randomly permuting all class labels $Y$ of the data set $Z$.

**Accuracy curves.** Central to our paper are accuracy plots. Here we plot the accuracy as computed on the training data over the randomness of the training data. Alternatively the plot of the accuracy as computed on the training data over the regularization parameter. In the paper we call such curves in the plots *accuracy over randomness curves* and *accuracy over regularization curves*. To generate the plots we keep everything fixed except the randomness of the training data or the regularization parameter.

**Monotony.** Let us assume that we successfully trained our model on the unperturbed data set $D_0 = Z$. This means that the accuracy over randomness curve starts in the left upper corner at some point close to 1. As we increase the level of randomness the training data $D_p$ becomes more complex and it is thus more difficult to learn, which means our algorithm will take longer to achieve the same training error. Thus, we expect that the accuracy drops. In other words the accuracy curve is strictly monotonically decreasing for increasing randomness. Further, if we increase the regularization parameter the model complexity drops. Thus we also expect that accuracy drops if the regularization of the model is increased. This shows that our assumption imply that the accuracy is strictly monotonically decreasing as a function of randomness and regularization.

Figure 1 shows the qualitative behavior of accuracy over randomness curves which follows the assumption we made.

**Real accuracy curves** To compare these idealized curves with accuracy curves of real data we computed the accuracy curves for different data sets, see Figure 2. In each subfigure we trained a neural network on either *mnist*[a], *cifar10*[b], and *patched-noise*[c] - a generated data set. For each curve we varied the $l_1$ regularization parameter. Furthermore, for each *randomness* value on the $x$-axis, the network was trained for five different randomization of the labels.

More details on the networks, the data sets and more plots can be found in the appendix.

## 3.3 THREE CRITERIA TO DETECT OVERFITTING

**Criterion 1 - Convexity of accuracy curve.** In Figure 1[a] three types of *accuracy curves* can be seen: dashed concave curves, dotted convex curves, and a full straight line. If the *accuracy curve* of our model is above the straight line, the model is able to learn noise. In other words the model overfits. Analogously, an accuracy curve below the straight line, shows underfitting. The model class has not enough capacity to learn the data.

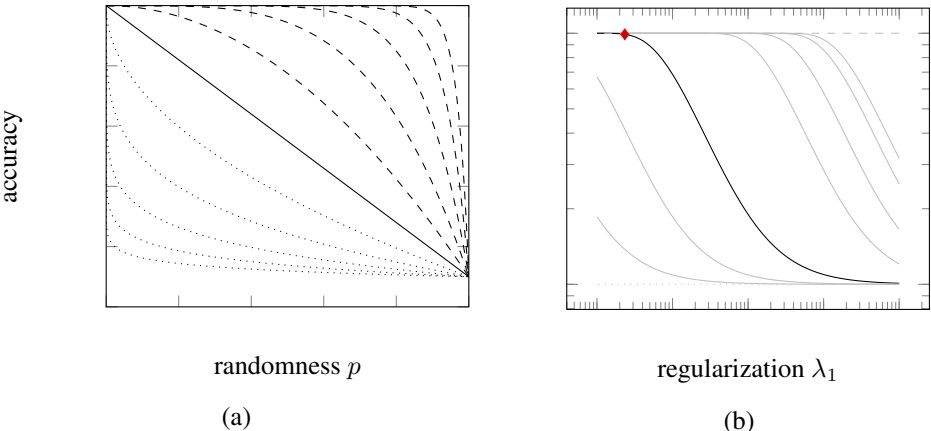

Figure 1: The figure shows the qualitative behavior of the *accuracy curves*. The dashed *accuracy over randomness curves*[a] depicts models which *overfit*, as the model has enough capacity to fit noise. The dotted *accuracy curves*[a] on the other hand show model which *underfit*. The shape of the *accuracy curves*[a] is controlled by a *regularization parameter* $\lambda_1$. Each curve in (b) depicts qualitatively a *accuracy over regularization curves*. The mark[b] depicts the point there the accuracy begins to descent.

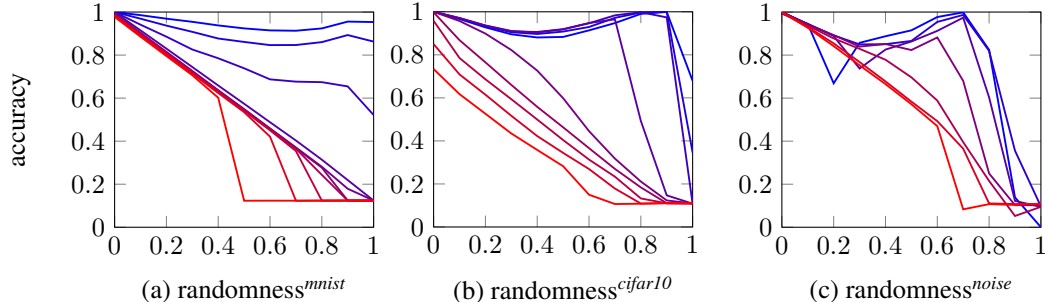

Figure 2: The figures show the accuracy curves for three different data sets. Each curve represents a different $l_1$-regularization. The curves start with no regularization, depicted in blue, to strong regularization depicted in red. As the regularization is increase the curve tend to pushed below the straight line, confirming our intuition.

*The model* overfits *if the accuracy computed on the true data set is close to one and the accuracy over randomness curve of the model is strictly concave. The model* underfits *if the accuracy curve is strictly convex.*

This criterion can be computed by measuring the squared distance of the accuracy curve to the straight line connecting the points $(0, 1)$ and $(1, \frac{1}{\text{number of classes}})$. So assuming the $r_1, ..., r_n$ parametrize the randomization of the labels, $a(r_i)$ denotes the accuracy at $r_i$ the criterion can be computed by:

$$\text{crit1} = \sum_i^n \left( a(r_i) - \left( (1 - r_i) + \frac{r_i}{\text{number of classes}} \right) \right)^2 \tag{5}$$

The criterion if met if crit1 is small.

**Criterion 2 - Steep decrease in accuracy.** Following our criterion 1 we want to determine if the accuracy curves are convex. Let us recall the accuracy curves are both strictly monotone decreasing in randomness and regularization. And that we are assuming that randomness and regularization are on a similar scale. If we look at the point in the upper left of Figure 1[(a)] we see that the curves are convex if the accuracy drops sharply as the randomness increases. As the accuracy curve is also monotone decreasing with increasing regularization we will also detect the convexity by a steep drop in accuracy as depicted by the marked point in the Figure 1[(b)].

*The model* overfits *if the accuracy on the training data is close to one and the accuracy over regularization curve (plotted in log-log space) is constant. Otherwise it underfits.*

This criterion can be detected by approximating the derivative $\text{crit2} = \frac{\partial}{\partial \lambda} a(\lambda)$ of the accuracy $a$ as the regularization parameter $\lambda$ increases. If the derivative becomes larger than a threshold, the optimal value is found.

**Criterion 3 - Two modes in margin histograms.** Finally we derive a criterion based on the margin histograms of the fully randomized training data. Looking again at 1 we see that the accuracy of the underfitting curves remains constant if we decrease the randomness just a tiny bit.

While training our model several things happen simultaneously. At starting time the network outputs random noise. If the parameter settings leads to a successful training, the model typically has a phase in which it outputs one class for all inputs. Looking at the margins of this phase, the distribution has two modes, one negative and a positive one containing the mass of $\frac{1}{\text{number of classes}}$ examples. Once we train further the two modes combine to one and the network starts to converge. Our *third criterion* looks for these two mode, because the accuracy will remain constant for a tiny decrease in randomness, as the two modes have to collapse before the accuracy can increase, we are thus in the underfitting regime.

*The model* overfits *if the margin histograms computed with fully random data* $D_1$ *respectively with true data* $D_0$ *are both positive. The model* underfits *if the margin histogram computed with fully randomized training data* $D_1$ *has two mode.*

This criterion can also be evaluated, by simply computing the quotient of positive margins to negative margins. So if $i$ denotes the index of a training sample, and $h_i$ its margin, then the criterion is computed by

$$\text{crit3} = \frac{\sum_i \chi_{\{h_i > 0\}}}{\sum_i \chi_{\{h_i < 0\}}} \tag{6}$$

where $\chi$ denotes the indicator function. The criterion is fulfilled if crit3 is close to $\frac{1}{\text{number of classes}}$.

## 4 EXPERIMENTAL SET UP

### 4.1 DATA SETS

Three data sets have been used to validate the criteria of this paper: *cifar10*, *minist* and *noise*.

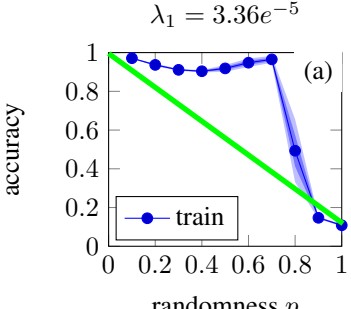 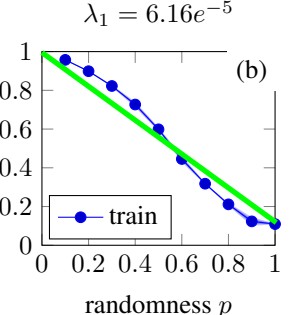 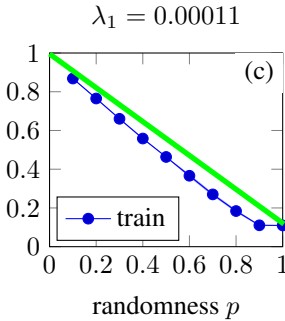

Figure 3: The plots show the accuracy curves of the network trained on *cifar10* over different degrees of randomness with increasing degree of $l_1$-regularization, after 19999 iterations. We select $\lambda$ such that the blue accuracy curve stays below the optimal green line and is convex. Following our convexity criterion we choose $\lambda_1^* = 0.00011$ as regularization factor.

**Cifar10.** We tested our criteria on *cifar-10*. This data set consists of 50000 images, equally divided in ten classes. As described above we generated random samples of the training data which we denote by *cifar-10^0.0*, *cifar-10^0.1*,...,*cifar-10^1.0*. A superscript stands for the fraction of randomized samples per class. So *cifar-10^0.0* stands for the original data. In *cifar-10^0.5* half of the labels of each class are randomly permuted while the other half remains fixed. Finally, in *cifar-10^1.0* all class labels are randomly permuted. Details on the architecture and training parameter can be found in the appendix.

**Mnist.** The data sets were created similar to *cifar10*.

**Noise.** This data set consists of 50000 randomly generated rgb noise, equally divided in ten classes. For each class a set of 100 random 5x5x3 patches were generated. Each image sampled 36 patches from the set of patches to build a 30x30x3 rgb image. In this way 50000 training and 50000 test images were generated. The randomization was similar to *cifar10*.

## 5 RESULTS

### 5.1 REGULARIZATION AND OVERFITTING

The capacity of a neural network is controlled by the number of parameters and (optionally) by a regularization term. In this section we show that the techniques of the paper can be used to tune the regularization parameter in order to avoid overfitting while still performing well.

**Convexity criterion.** For the *convexity criterion*, we compute the accuracy curves over increasingly randomized data for different accuracy parameters as shown in Figure 3. We expect that our algorithm achieves zero training error for true data. Furthermore for $p$-percent of randomized training data we expect the algorithm to achieve an error of $p$-percent. In other word we expect the algorithm to stay below a straight line starting at 1 for *data^0.0* and going down to 1/*number of classes* for *data^1.0*. Let us call this line the *optimal line*. We pick the smallest $\lambda$ for which the training error stays below the *optimal line* and is convex.

$l_1$**-regularization.** In these set of experiments we varied the *regularization factor* of the $l_1$ regularization of our *Alexnet*-type network on *cifar10*. To get more reliable results we run the experiments for five different random samples. From the samples we computed the mean and the standard deviation, which we indicated in the plots by shading. Our experiments show that all criteria lead to a similar regularization factor $\lambda_*$.

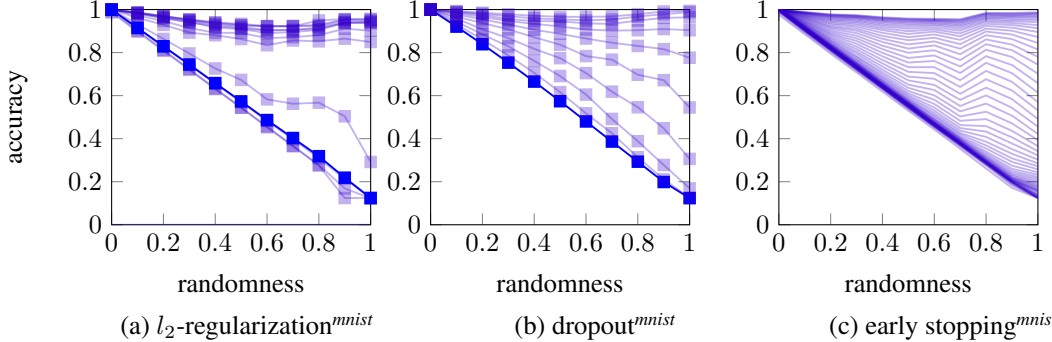

Figure 4: For (a) an *Alexnet*-type network was trained on *mnist* with varying degrees of randomness. With increasing regularization the curves approach the straight line, from which the optimal regularization parameter $0.0023$ can be determined. In (b) the same network was trained on *mnist* with varying dropout rate and no other regularization. With decreasing drop out rates the curves approach the straight line. In the setting of the experiment a drop out rate of $0.1$ is optimal. For (c) we trained an *Alexnet*-type neural network with $l_2$ regularization with parameter $0.0012$, and computed the accuracy curves for the steps $1000, 2000, ..., 60000$. For $1000$ training steps the curve is the lowest, for $60000$ training steps the curve approaches the constant $1$.

In Figure 3 the *optimal line* is depicted in green. According to our criterion we choose $\lambda_1 = 0.00011$. Both criteria (C2) and (C3) lead to a similar regularization parameter, details can be found in appendix B.1.

$l_2$**-regularization.** In these set of experiments we varied the *regularization factor* of the $l_2$ regularization of a (different) *Alexnet*-type network on *mnist*. Again we run the experiments for five different random samples and computed the mean and the standard deviation, which we indicated in the plots by shading. The resulting accuracy curves are shown in Figure[(a)] 4, the resulting parameter is $\lambda_2 = 0.0023$.

**Dropout.** Dropout is another way to regularize a neural network. To analyze its effect on overfitting we trained a *Alexnet*-type network on *mnist* with different probabilities of drop out ranging from $0.1$ to $1.0$. The resulting accuracy curves are shown in Figure[(b)] 4, in the setting of the experiment the optimal drop out value is $0.1$.

**Early stopping.** Early stopping follows the rational less training time leads to better generalization properties. In Figure[(c)] 4 we can observe this rational. The more we train the network the more the network overfits. The curves are a bit more bumpy as we trained the models only once.

## 5.2 COMPARISON WITH TEST SET

In this section we compare the training accuracy with the test accuracy. The plots of Figure show nine panels with different regularization factors. In each panel the accuracy as computed on the training data is plotted in blue, and the accuracy of the test data in red. So each blue dot represents the accuracy of the different training sets *cifar10^{0.0}* ... *cifar10^{1.0}*. Each red point is computed for the same test set.

With no regularization the model highly overfits. The model learns randomized data rather well as shown in $\lambda_1 = 0.0$ in Figure 7. We further observe that it is easier for the network to learn more random data. With no $l_1$-regularization the accuracy curves decreases with increasing randomness in the data and then starts to increase again. We attribute this to correlation in the data set which make the training data more complex for lower noise levels. Higher noise levels destroy these correlations and the data complexity of the data reduces. Recall that the small animal classes and also the car / truck class are correlated in *cifar10*. Finally, we note that the variance for learning entirely random data is very high.

As the regularization parameter increases the blue accuracy show that the network is less able to learn random data. For $\lambda_1 = 0.00011$ the curve is convex, showing the optimal regularization parameter. In $\lambda_1 > 0.00011$ the model underfits. Looking again at $\lambda_1 = 0.00011$ we see that the model is able to learn from noisy data with lots of label noise. This confirms that $l_1$-regularization is a good parameter to adjust the model complexity.

## 5.3 EARLY STOPPING

Plots similar to Figure 5 can be used to analyze *early stopping* and overfitting. Due to lack of space we will only describe the results verbally. Early in the training, at 19999 steps, we see that almost all curve are convex, hence the models underfit. Once we train the network to 59999 iterations, the model trained without any regularization begins to overfit with the others still underfitting. Training the networks further the more and more models begin to overfit. Flipping through the plots in the appendix illustrates this nicely.

## 5.4 FILTER SIZE AND NETWORK DEPTH

We also looked at models with different filter sizes in the first convolutional layer. We trained several networks with filter sizes starting from $2 \times 2$ to $9 \times 9$ and a regularization parameter of $\lambda_1 = 0.00011$. We observed that all networks showed underfitting, revealed by the convexity of the accuracy over randomness curves. This hints that $l_1$ regularization of the kernel weights is more important to overfitting than the number of parameters. Experiments with different network depths showed a similar behavior.

## 6 DISCUSSION AND CONCLUSION

In the paper we measure the capacity of a neural network by injecting different noise levels in the training data. The criteria we introduced in the paper are based on the assumption that the network should only be able to achieve a training accuracy corresponding to the injected noise level. This advances previous method in the neural network setting as they rely on either a hold out set, heuristics, or generalization theory. All of which are not mature enough to detect overfitting at present. In our experiments we saw that the hyper parameters fall in two classes, one which has no effect on overfitting (kernel size) and another which controls overfitting (regularization factor, number of iterations). In other experiments on *mnist* and *cifar10* we observed the dominance of $l_1$ regularization for overfitting, while structural parameters such as network width, depth did not had an effect.

The *convexity criterion* is the most reliable, as outliers and high variance are easily detected. On the downside it requires the most training runs. The *steep decrease criterion* only requires to train the model on the real data and and the fully random data. It can be used to narrow the parameter range. On the down side correlation between the classes are not easily detected by the *steep decrease criterion*. The *mode criterion*, is the most easiest to use as only the totally randomized training data is used. On the downside the margin plots are not always easy to interpret. Either the entire margin is positive, then the model clearly overfits, or two modes are observed in the plots, then the model clearly underfits. Yet most of the time, the margin is somewhere in between, which makes it hard to make a judgment based on the margin histograms alone.

Let us put criteria (C2) and (C3) in perspective. Criterion (C2) comes close to what has been done before. We basically train a network on true and randomly shuffled labels, and analyze the attained accuracies. An analysis of the margin histograms for networks trained on true labels and random labels has been explored before. For example in Bartlett et al. (2017) margin histograms are used to conclude that *regularization only seems to bring minor benefits to test error*, Liang et al. (2017) use the margin histograms of networks trained on fully randomized labels and true labels to discuss normalization effects. Our contribution is to show that the regularization parameter can be set such that network does train on true labels, but is unable to do so for random labels. Both criteria are able to note this effect.

All criteria can be numerically evaluated and put into an automated parameter search. At present it seems that the number of parameters do not contribute to overfitting. Thus to use the criteria of this

paper one would proceed in two steps: search for an architecture which achieves zero training error, and then reducing the complexity of the model by regularizing it such that it does not overfit. So the additional burden is not that much .

Analyzing neural networks with randomized training data has been done before (Zhang et al. (2017)). In the paper the authors show that a neural network is able to train random labels, and they note that regularization ... *is neither necessary nor by itself sufficient for controlling generalization error.* In the paper we argued that $l_1$-normalization of the kernel weights is a good measure to control the capacity of a network. In the experiment we saw that adjusting $l_1$-normalization leads to models which do not overfit and hence we expect them to generalize better. Using an $l_1$ regularization (the LASSO) is one of the popular choices for regularization. The rational is typically to enforce sparsity of the network weights. Our Lemma 3.1.1 adds another reason to the list why it might be a good choice for convolutional networks.

We want to highlight another unexpected illustrative result. By tuning the hyper parameter to pass our overfitting tests, we see that the test accuracy of the model is much higher than the training accuracy. This shows that our criteria can also be used to learn from noisy data and that a generalization gap does not need to be a bad thing.

Although the paper focused on neural networks the methods can be applied for other machine learning algorithms as well. For example it would be interesting to apply our criteria for a systematic architecture search. Another line of research could investigate whether the criteria make adversarial attacks more difficult.

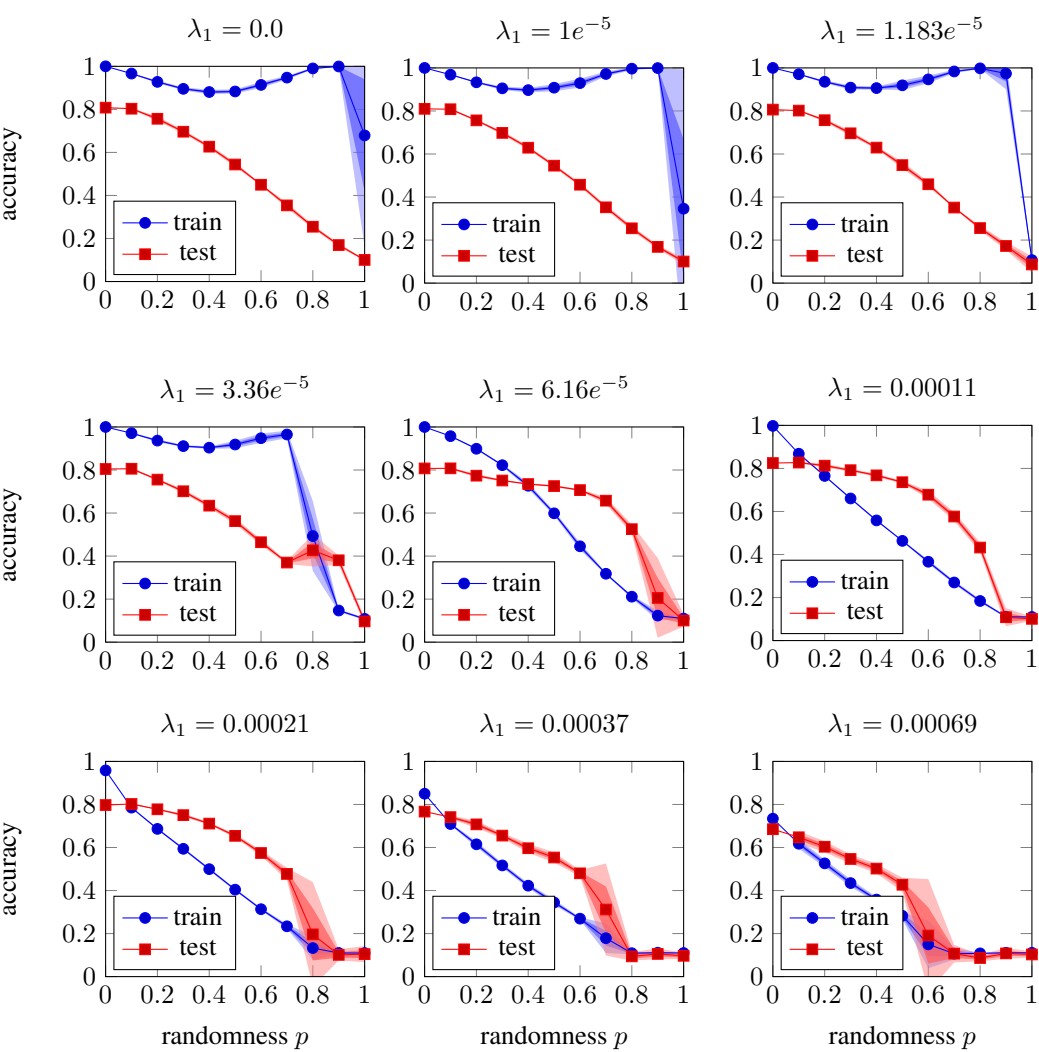

Figure 5: The plots shows the accuracy of the network trained on *cifar10* over different degrees of randomness with increasing degree of $l_1$-regularization. The network trained for 199999 iterations. For the error curves five different samples were sampled for each data point. The network was evaluated on the training set (depicted in blue) and on the test set (depicted in red). We observe that the model does not overfit for $\lambda = 0.00011$. Furthermore, we note that with this choice of $\lambda$ the model is able to learn from noise data, as the red curve is clearly above the green noise level curve.

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

# A MATH

## A.1 ADDITIONAL MATH BACKGROUND

**Matrix norms.** A matrix $A\colon V \to W$ can be viewed as a linear operator between two normed spaces $(V, ||\cdot||_p)$ and $(W, ||\cdot||_q)$. We equip these normed spaces with a $p$-norms. So for $x \in V$ we set $||x||_p = (\sum_i |x_i|^p)^{\frac{1}{p}}$ and for $y \in W$ we set $||y||_q = (\sum_i |y_i|^q)^{\frac{1}{q}}$. These vector space norms induce a matrix norm for $A$:

$$||A||_{p\to q} := \frac{||Ax||_q}{||x||_p}$$

Special cases of this norm include the *spectral norm* $||A||_\sigma = ||A||_{2\to 2}$, $||A||_{1\to 1} = \max_{1\le j\le n} \sum_{i=1}^m |a_i j|$ and $||A||_{\infty\to\infty} = \max_{1\le j\le m} \sum_{i=1}^n |a_i j|$. In the paper we use the following fact:

$$||A||_{2\to 2} \le \sqrt{||A||_{1\to 1}||A||_{\infty\to\infty}} \tag{7}$$

A definition of these norms can be found in books about matrix analysis see for example §2.3.1 Golub & Van Loan (1996) for a definition of the $||A||_{p\to q}$ norm (in a slightly different notation). Equation (7) can be found in Corollary 2.3.2 of the same reference.

**Generalized mean.** For a non zero real number $p$ and positive reals $x_1, \ldots, x_n$ we define the *generalized mean* by

$$M_p(x_1, \ldots, x_n) = \left( \frac{1}{n} \sum_{i=1}^n x_i^p \right)^{\frac{1}{p}}. \tag{8}$$

We will use the following inequality which holds true for all real $p < q$ and positive $x$

$$M_p(x_1, \ldots, x_n) \le M_q(x_1, \ldots, x_n). \tag{9}$$

**Theorem A.1.1** (Bartlett et al. (2017) Theorem 1.1). *Let nonlinearities $(\phi_1, \ldots, \phi_L)$ and reference matrices $(M_1, \ldots, M_L)$ be given with $\sigma_i$ is $\rho_i$-Lipschitz and $\sigma_i(0) = 0$. Then for $(x, y), (x_1, y_1), \ldots, (x_n, y_n)$ drawn iid from some probability distribution over $\mathbf{R}^{d_0} \times \{1, \ldots, k\}$, with probability at least $1 - \delta$ over $((x_i, y_i))_{i=1}^n$, every margin $\gamma > 0$ and admissible network $f\colon \mathbf{R}^d \to \mathbf{R}^k$ with weight matrices $A_1, \ldots, A_L$ satisfy*

$$\text{Prob}\left[ \arg\max_i f(x)_i \ne y \right] \le \hat{\mathcal{R}}_\gamma(f) + \tilde{O}\left( \frac{R_A ||X||_2}{\gamma n} + \sqrt{\frac{\ln(1/\delta)}{n}} \right) \tag{10}$$

*where $\hat{\mathcal{R}}_\gamma(f)$ is the empirical ramp risk and $||X||_2 = \sqrt{\sum_i ||x_i||_2^2}$*

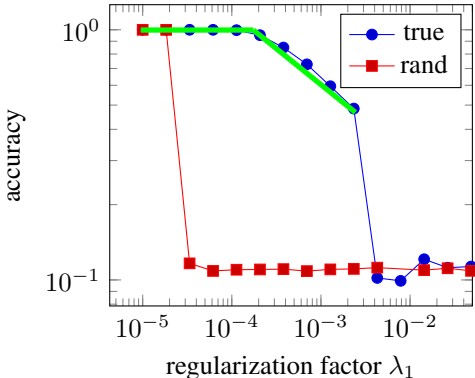

Figure 6: In our *steep descent criterion* we propose to detect the first change point at which the accuracy is not constant anymore but falls linearly. In the figure this is depicted by the green curves: around $10^{-4}$ the accuracy begins to fall. As a measure of how much the net learned about the data, we also provide the accuracy curves for random data. We conclude from the gap between the red and the blue curve that the net learned something meaningful about the data, instead of just memorizing the data.

## B    COMPARISON OF THE CRITERIA

In this section we empirically show that criteria (C2) and (C3) lead to similar conclusions than criterion (C1).

### B.1    REGULARIZATION FACTOR

**Regularization.**    In these set of experiments we varied the *regularization factor* of the $l_1$ regularization of our *Alexnet*-type network. To get more reliable results we run the experiments for five different random samples. From the samples we computed the mean and the standard deviation, which we indicated in the plots by shading. The following experiments show that all criteria lead to a similar regularization factor $\lambda_*$.

**Steep decrease criterion.**    To test our *steep decrease criterion* we computed the accuracy over regularization curve. As the regularization increases we expect the accuracy to drop. This is shown in Figure 6. Following our criterion the point of interest $\lambda^*$ occurs, at which the curve is not constant anymore. This occurs around $\lambda_1 = 0.0001$.

**Mode criterion.**    To test our *mode criterion* we computed the margin histograms of *cifar-10[1.0]* after training. As the regularization increases we expect the distribution to split up in two modes. This can be seen in Figure 7. Following our criterion the point of interest occurs around $\lambda_1 = 0.00011$.

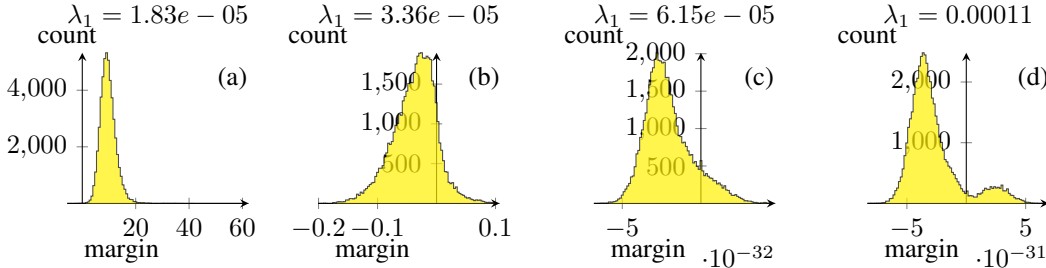

Figure 7: The plots[(a)-(d)] show the margin histograms as trained in *cifar10[1.0]* for an increasing regularization factor. According to our third criterion we choose the $\lambda = 0.00011$, as here[(d)] the margin distribution developed a second mode for the first time.

## C   ARCHITECTURE DETAILS

**Architecture and training parameter.**   Figure 8 shows a sketch of the architecture used in most experiments. We start with $5 \times 5$ convolutional filters, followed by $3 \times 3$-convolutional filters and two fully connected layers. In all layers the linear part is followed by a *relu* nonlinearity. We did not use a bias. In addition we apply three non overlapping $2 \times 2$-max-pooling. Further we used drop-out between the first and second fully connected layer. Everything was coded in *Tensorflow*, with SGD, a fixed learning rate of 0.01, a batch size of 32 and $l_1$ normalization of all weights. The networks were trained for 199999 steps. The input images were normalized to zero mean and standard deviation using one of *tensorflows* build in function. Additionally, we used some standard data augmentation.

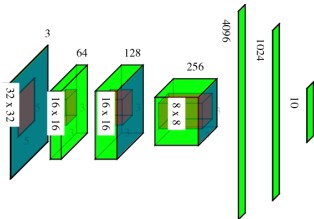

Figure 8: The figure shows a sketch of the networks used for most experiments on *cifar10*.

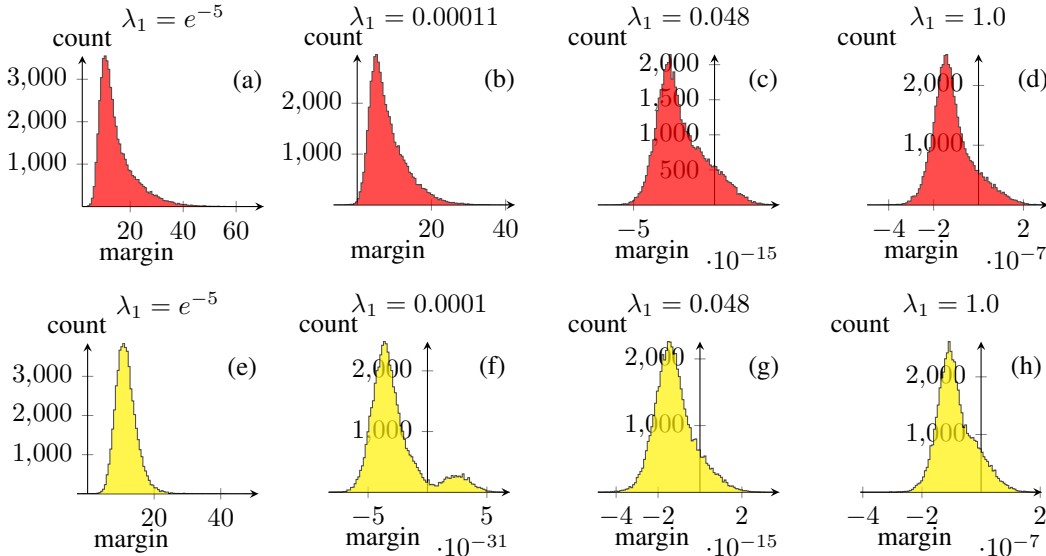

Figure 9: The plots show different $l_1$-regularizations for *cifar10*[(a)-(d)] and *cifar10-random*[(e)-(h)]. Beginning in (a) (and (f)) the regularizations are $e^{-5}, 0.0001, 0.048, 1.0$. We note that for (a) and (e) the model overfits, for (b) and (f) it is just right, and for (c) and (g) it underfits.

## D    ADDITIONAL PLOTS

### D.1    $l_1$ REGULARIZATION

Here we provide additional plots of our $l_1$ regularization experiments, showing that all criterion have their uses. Figure shows how we would detect overfitting with the margin based criterion. Let us recall that a positive margin corresponds to a correct classification and a negative margin corresponds to an incorrect classification. In (a) and (e) of Figure D.1 the model clearly overfits, as it is able to learn random data[(e)] and true data[(a)]. In (c) and (g) of Figure D.1, we clearly see underfitting the model neither able to learn random data nor true data. Based on this observation we would select $\lambda = 0.0001$ as our regularization parameter.

### D.2    EARLY STOPPING

Here we report similar plot for our early stopping experiments. Flipping through the plots we see that initial the regularization factor does not matter at 19999 steps all curves are convex. At later iterations the models begin to memorize the data.

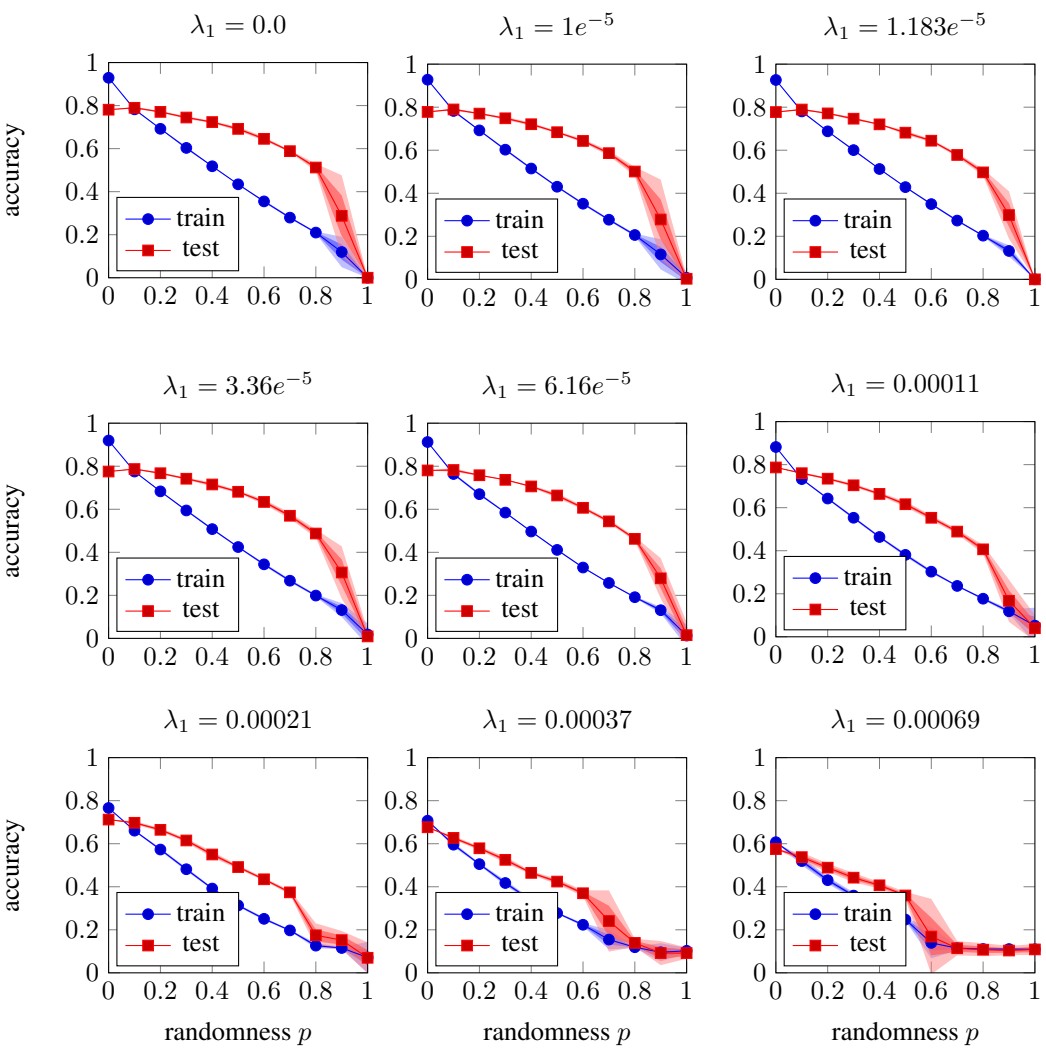

Figure 10: The plots shows the accuracy of the network trained on *cifar10* over different degrees of randomness with increasing degree of $l_1$-regularization. The network trained for 019999 iterations. For the error curves five different samples were sampled for each data point. The network was evaluated on the training set (depicted in blue) and on the test set (depicted in red).

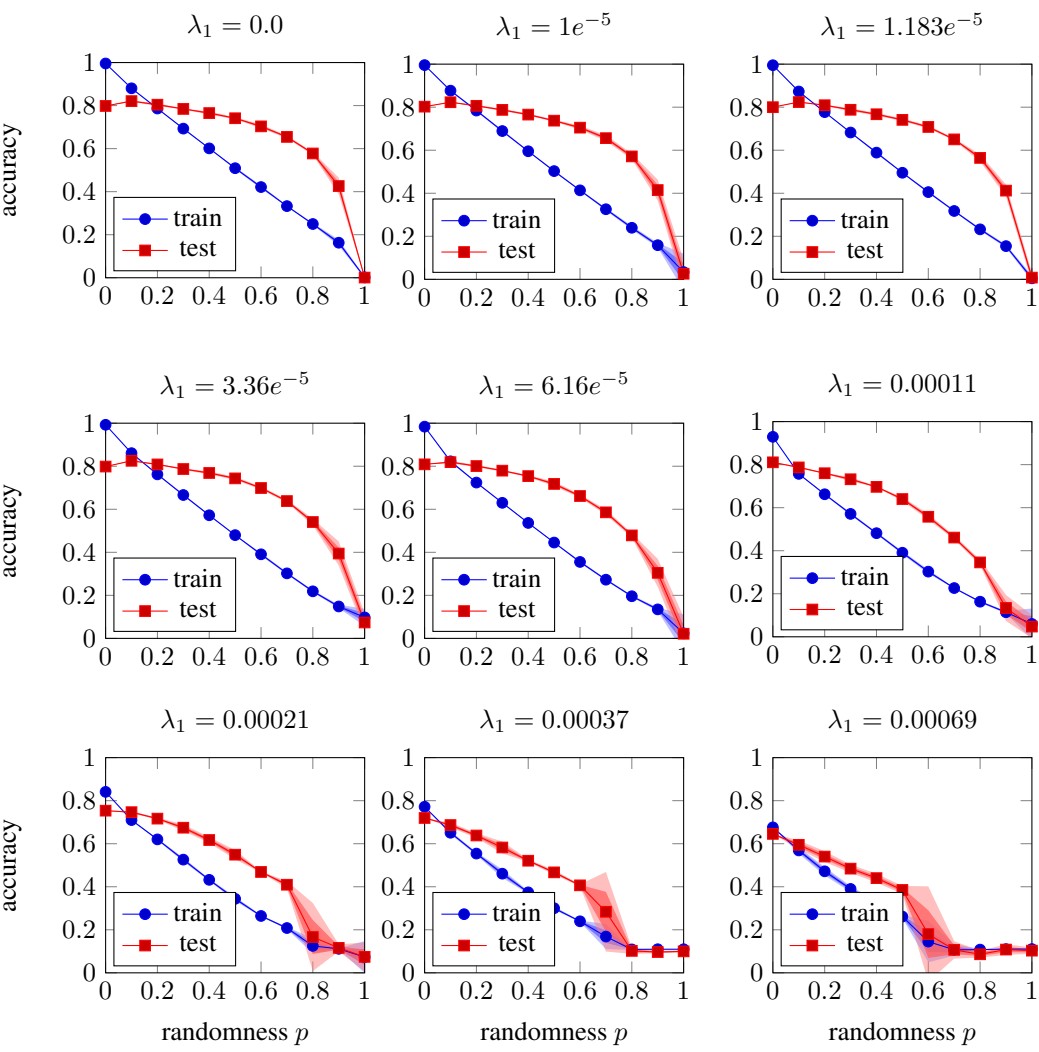

Figure 11: The plots shows the accuracy of the network trained on *cifar10* over different degrees of randomness with increasing degree of $l_1$-regularization. The network trained for 039999 iterations. For the error curves five different samples were sampled for each data point. The network was evaluated on the training set (depicted in blue) and on the test set (depicted in red).

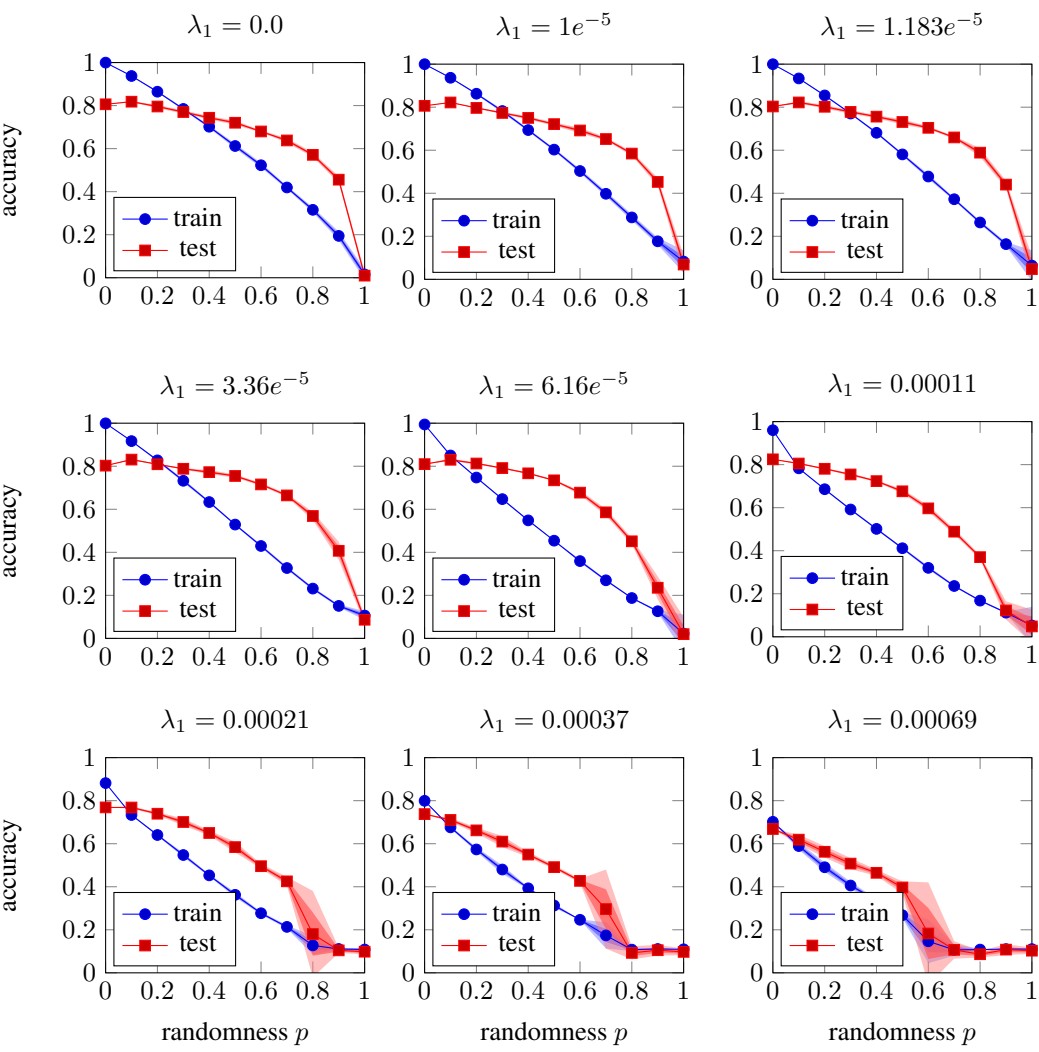

Figure 12: The plots shows the accuracy of the network trained on *cifar10* over different degrees of randomness with increasing degree of $l_1$-regularization. The network trained for 059999 iterations. For the error curves five different samples were sampled for each data point. The network was evaluated on the training set (depicted in blue) and on the test set (depicted in red).

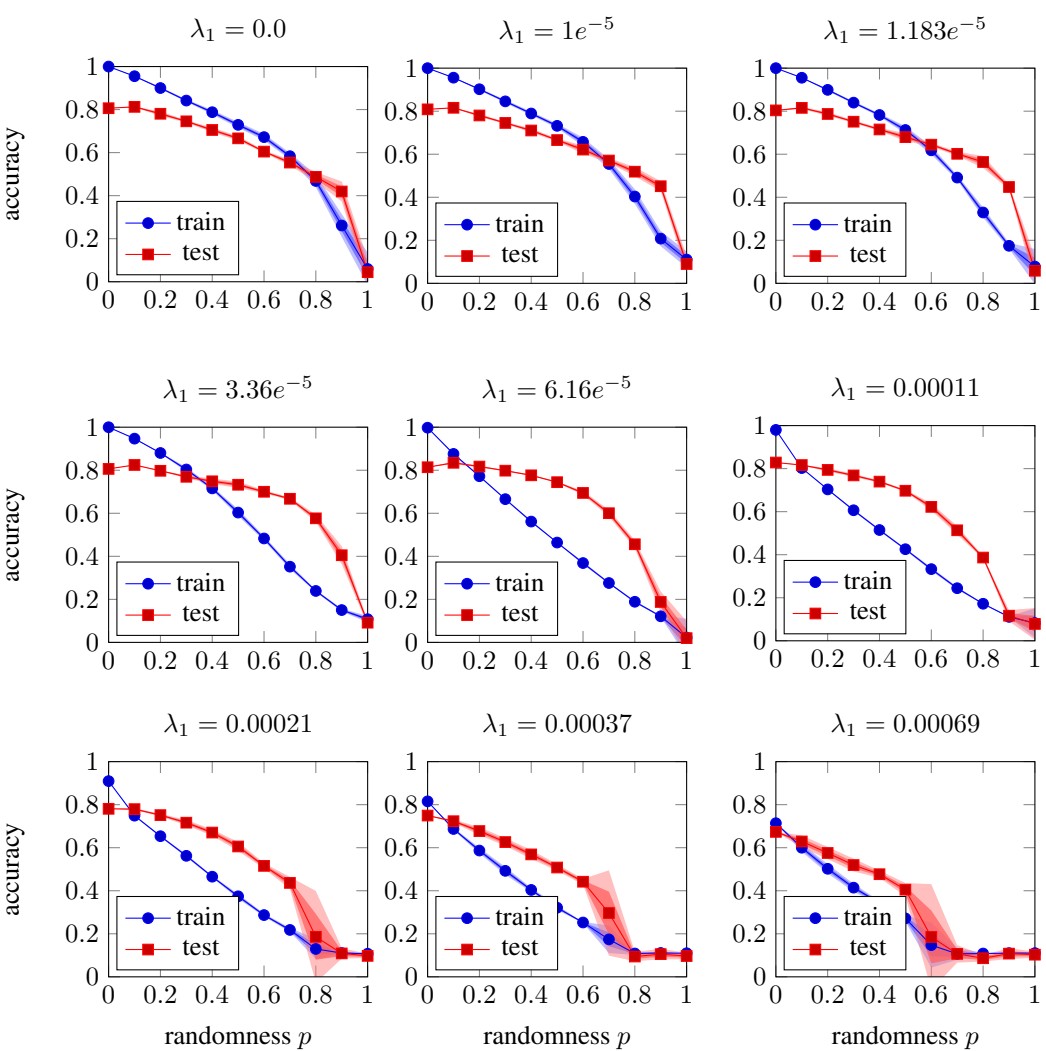

Figure 13: The plots shows the accuracy of the network trained on *cifar10* over different degrees of randomness with increasing degree of $l_1$-regularization. The network trained for 079999 iterations. For the error curves five different samples were sampled for each data point. The network was evaluated on the training set (depicted in blue) and on the test set (depicted in red).

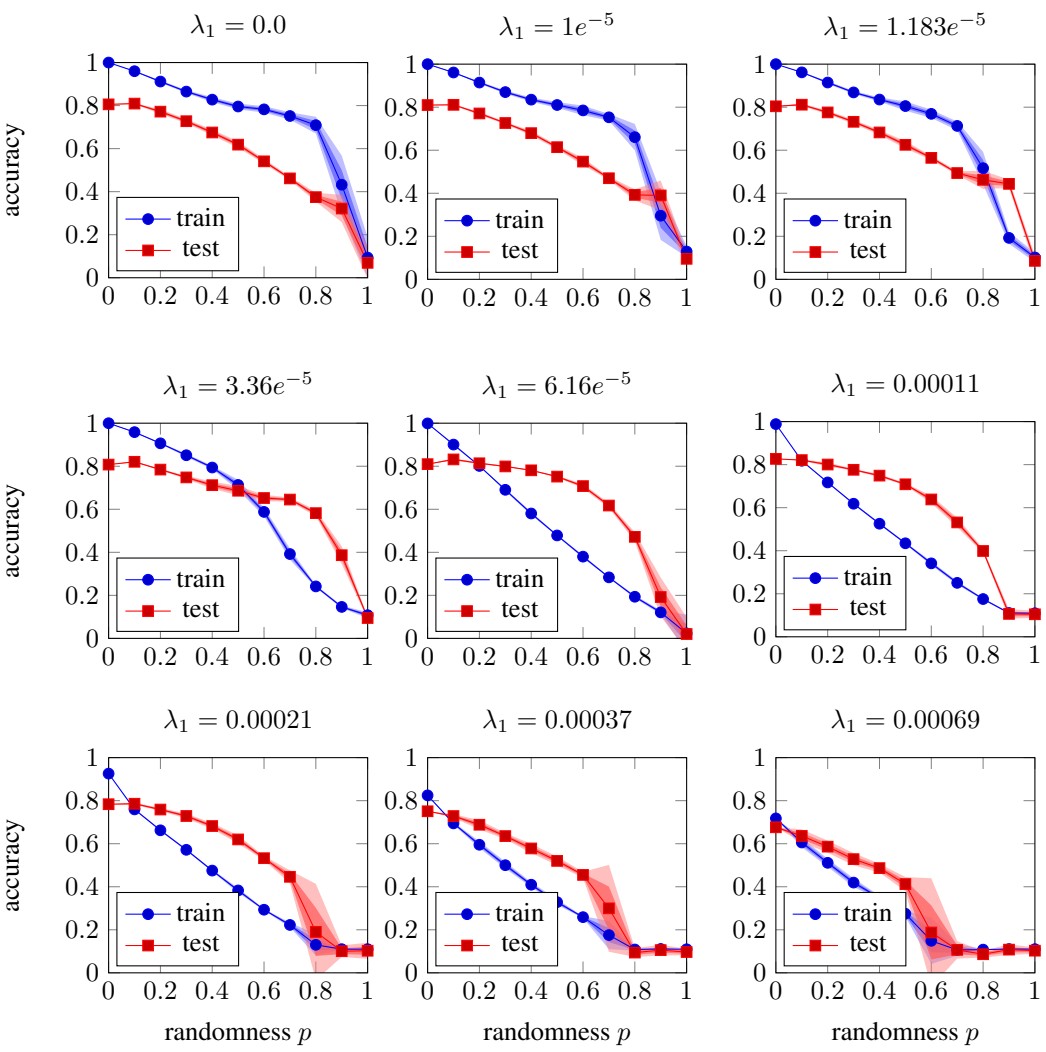

Figure 14: The plots shows the accuracy of the network trained on *cifar10* over different degrees of randomness with increasing degree of $l_1$-regularization. The network trained for 099999 iterations. For the error curves five different samples were sampled for each data point. The network was evaluated on the training set (depicted in blue) and on the test set (depicted in red).

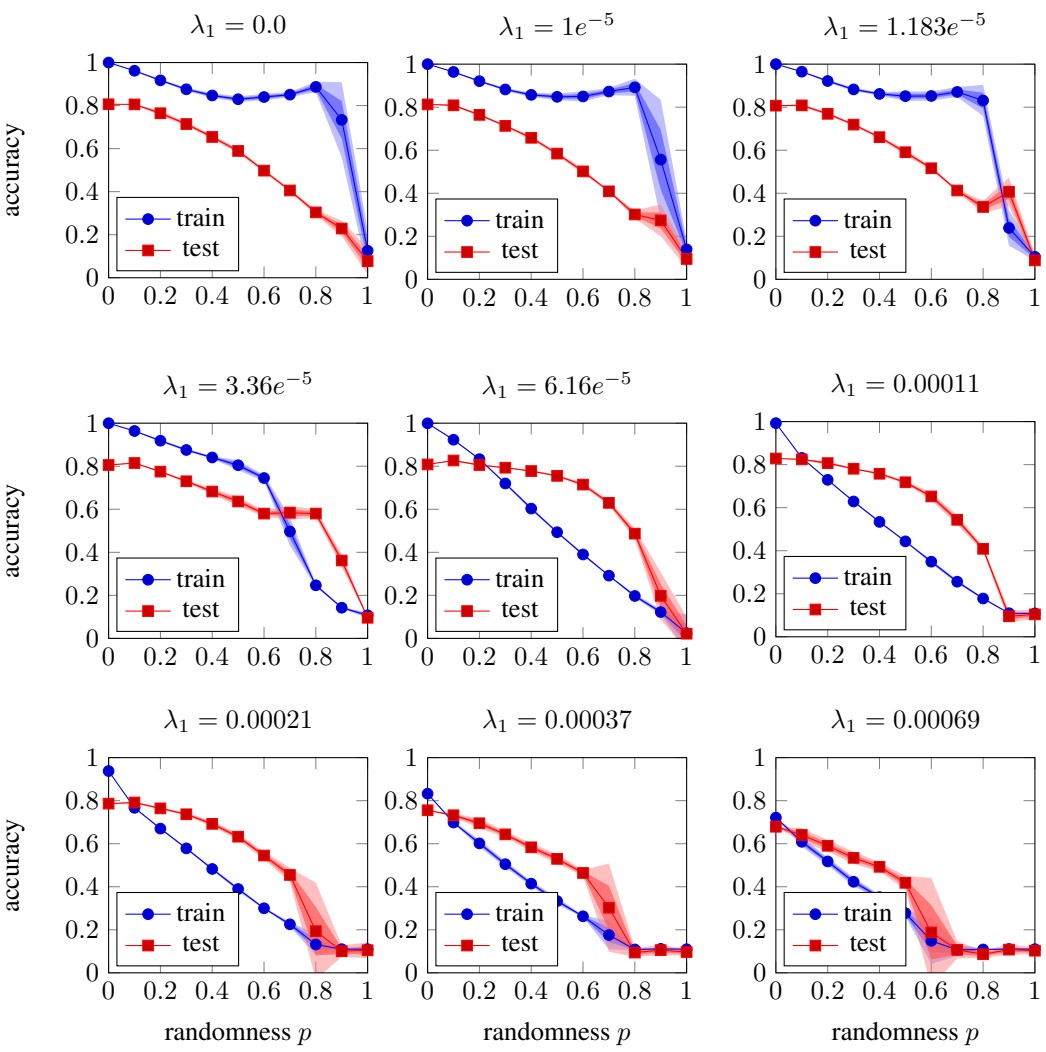

Figure 15: The plots shows the accuracy of the network trained on *cifar10* over different degrees of randomness with increasing degree of $l_1$-regularization. The network trained for 119999 iterations. For the error curves five different samples were sampled for each data point. The network was evaluated on the training set (depicted in blue) and on the test set (depicted in red).

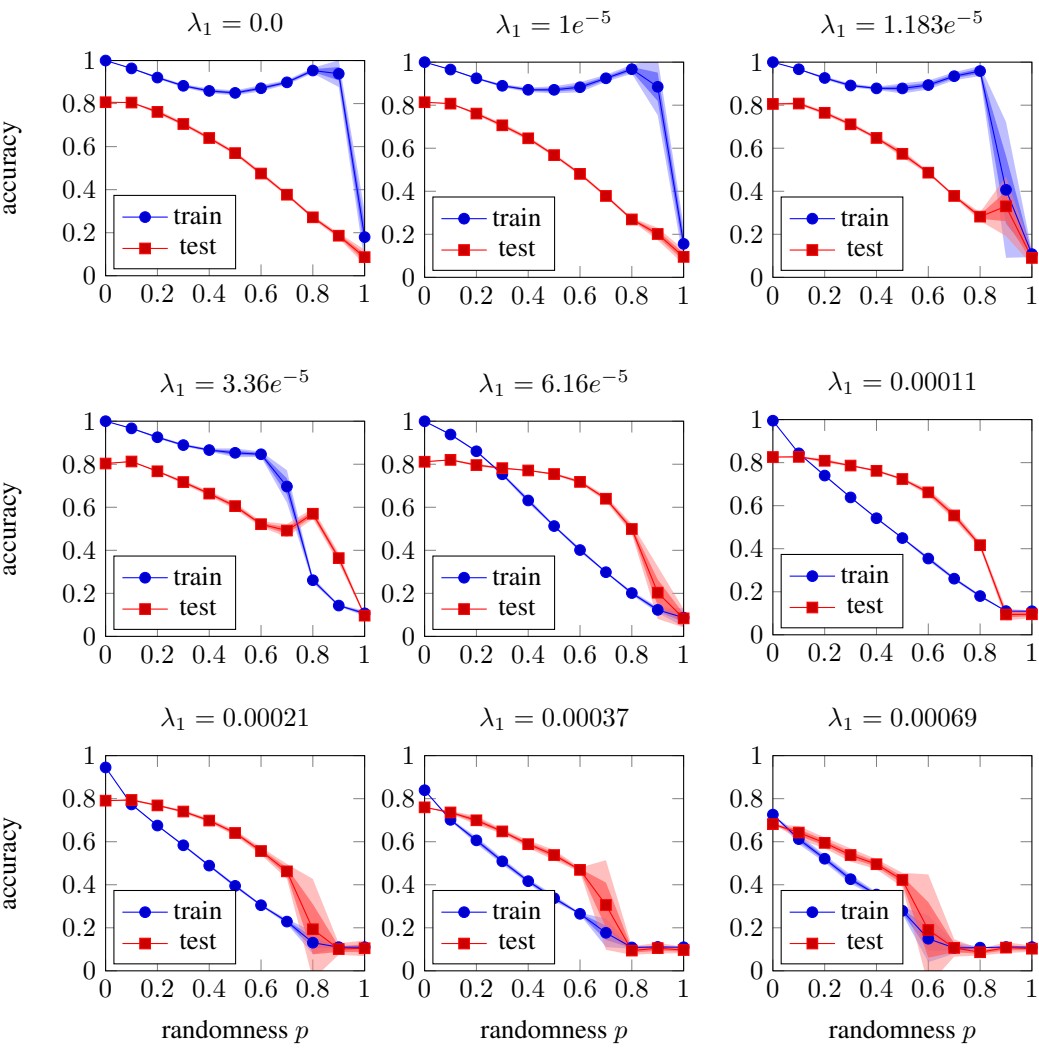

Figure 16: The plots shows the accuracy of the network trained on *cifar10* over different degrees of randomness with increasing degree of $l_1$-regularization. The network trained for 139999 iterations. For the error curves five different samples were sampled for each data point. The network was evaluated on the training set (depicted in blue) and on the test set (depicted in red).

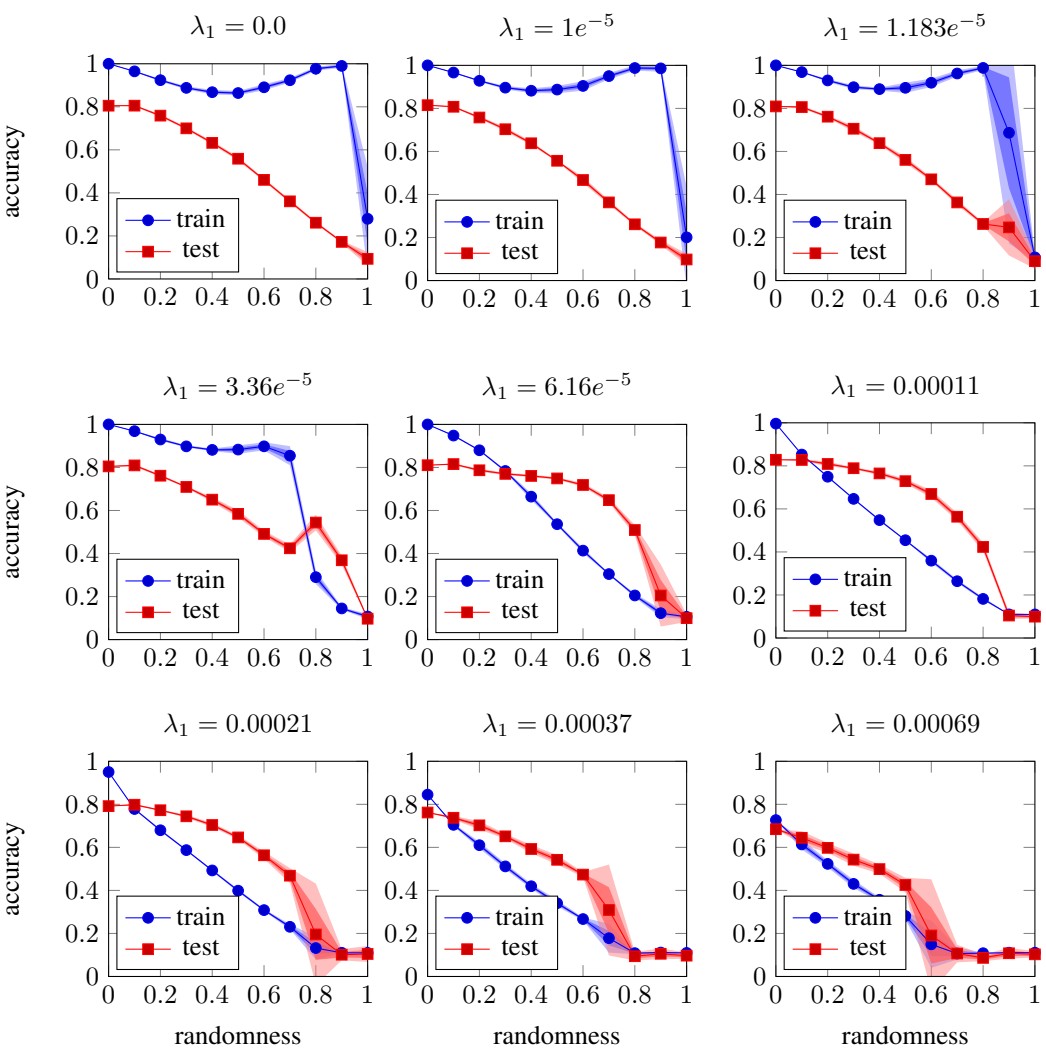

Figure 17: The plots shows the accuracy of the network trained on *cifar10* over different degrees of randomness with increasing degree of $l_1$-regularization. The network trained for 159999 iterations. For the error curves five different samples were sampled for each data point. The network was evaluated on the training set (depicted in blue) and on the test set (depicted in red).

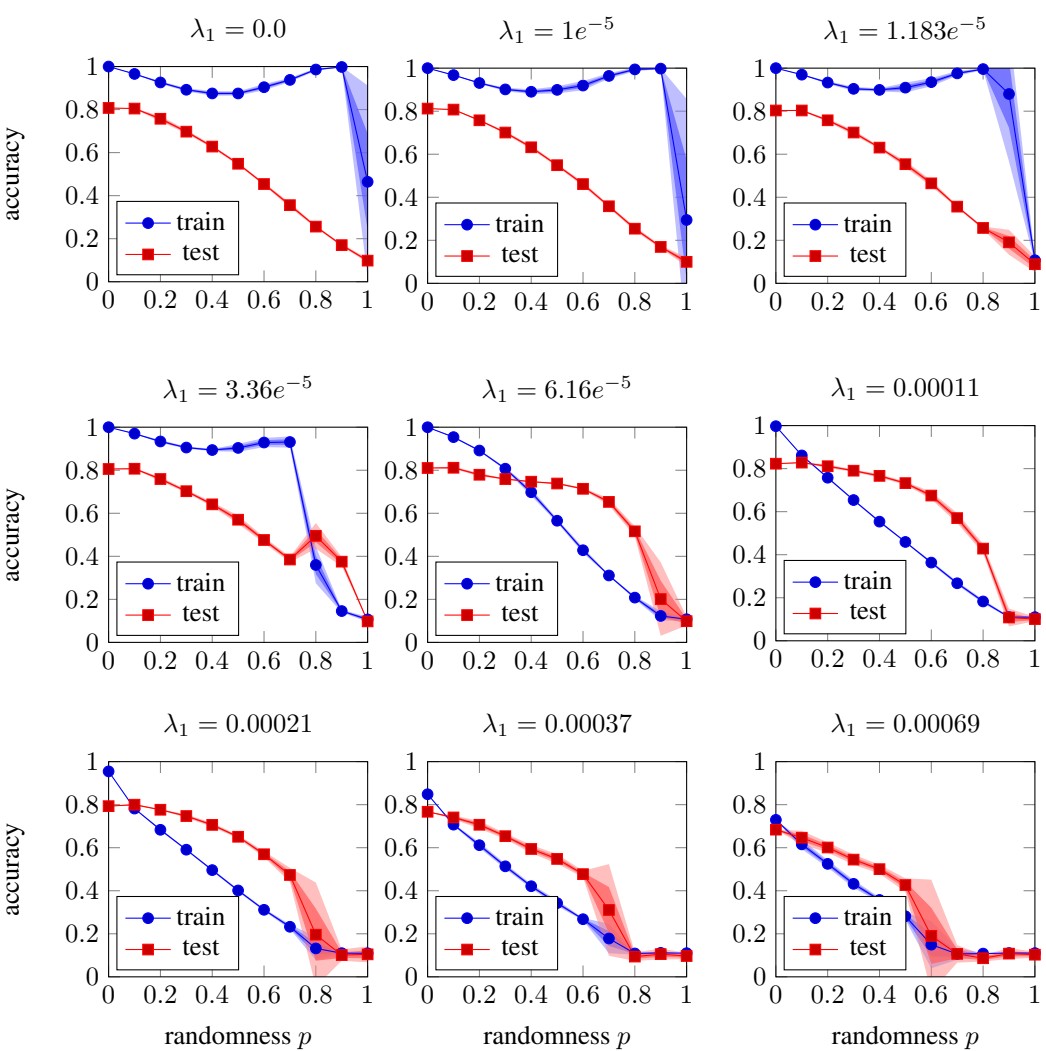

Figure 18: The plots shows the accuracy of the network trained on *cifar10* over different degrees of randomness with increasing degree of $l_1$-regularization. The network trained for 179999 iterations. For the error curves five different samples were sampled for each data point. The network was evaluated on the training set (depicted in blue) and on the test set (depicted in red).

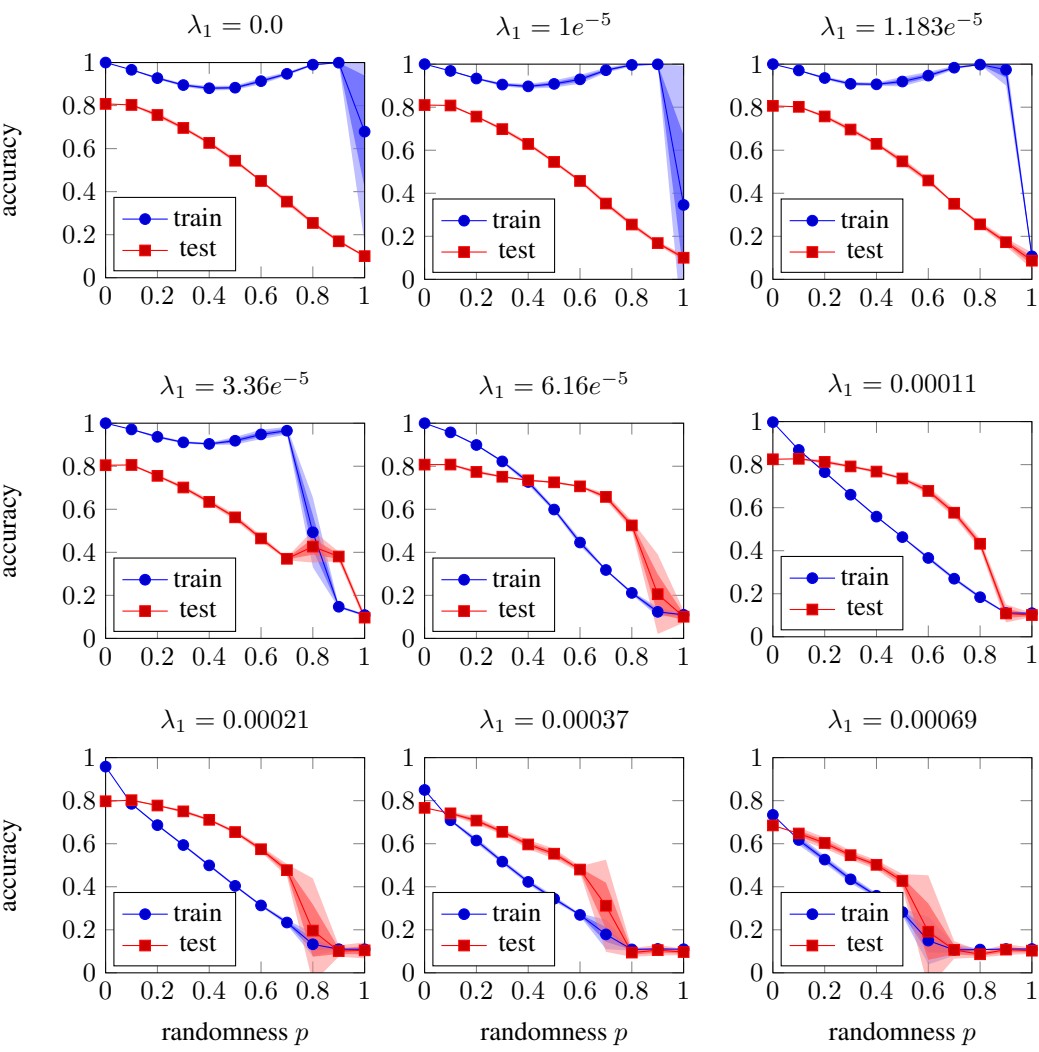

Figure 19: The plots shows the accuracy of the network trained on *cifar10* over different degrees of randomness with increasing degree of $l_1$-regularization. The network trained for 199999 iterations. For the error curves five different samples were sampled for each data point. The network was evaluated on the training set (depicted in blue) and on the test set (depicted in red).

