# OpenReview forum: "Overfitting Detection of Deep Neural Networks without a Hold Out Set"
_ICLR.cc/2019/Conference_

### Official Review · AnonReviewer3 · 2018-11-02
**Potential overfitting criteria remain vague and were not properly validated**

**Rating:** 3
**Confidence:** 4

**Review:**

Overview:
The authors aim at finding and investigating criteria that allow to determine whether a deep (convolutional) model overfits the training data without using a hold-out data set.
Instead of using a hold-out set they propose to randomly flip the labels of certain amounts of training data and inspect the corresponding 'accuracy vs. randomization‘ curves. They propose three potential criteria based on the curves for determining when a model overfits and use those to determine the smallest l1-regularization parameter value that does not overfit.
I have several issues with this work. Foremost, the presented criteria are actually not real criteria (expect maybe C1) but rather general guidelines to visually inspect 'accuracy over randomization‘ curves. The criteria remain very vague and seem be to applicable mainly to the evaluated data set (e.g. what defines a ’steep decrease’?). Because of that, the experimental evaluation remains vague as well, as the criteria are tested on one data set by visual inspection. Additionally, only one type of regularization was assumed, namely l1-regularization, though other types are arguably more common in the deep (convolutional) learning literature.
Overall, I think this paper is not fit for publication, because the contributions of the paper seem very vague and are neither thoroughly defined nor tested.


Detailed remarks:

General:
A proper definition or at least a somewhat better notion of overfitting would have benefitted the paper. In the current version, you seem to define overfitting on-the-fly while defining your criteria.

You mention complexity of data and model several times in the paper but never define what you mean by that.


Detailed:
Page 3, last paragraph: Why did you not use bias terms in your model?

Page 4, Assumption.
- What do you mean by the data being independent? Independent and identically distributed?
- "As in that case correlation in the data can be destroyed by the introduction of randomness making the data easier to learn.“ What do you mean by "easier to learn"? Better generalization? Better training error?
- I don’t understand the assumptions. You state that the regularization parameter should decrease complexity of the model. Is that an assumption? And how do you use that later?
- What does "similar scale“ mean?

Page 4, Monotony.
- You state two assumptions or claims, 'the accuracy curve is strictly monotonically decreasing for increasing randomness‘ and 'we also expect that accuracy drops if the regularization of the model is increased’, and then state that 'This shows that the accuracy is strictly monotonically decreasing as a function of randomness and regularization.‘ Although you didn’t show anything but only state assumptions or claims (which may be reasonable but are not backed up here).
I actually don’t understand the purpose of this paragraph.

- Section 3.3 is confusing to me. What you actually do here is you present 3 different general criteria that could potentially detect overfitting on label-randomized  training sets. But you state it as if those measures are actually correct, which you didn’t show yet.

My main concern here, besides the motivations that I did not fully understand (s.b.), is the lack of measurable criteria. While for criterion 1 you define overfitting as 'above the diagonal line‘ and underfitting as ‚below the line‘, which is at least measurable depending on sample density of the randomization, such criteria are missing for C2 and C3.       Instead, you present vague of ’sharp drops’ and two modes but do not present rigorous definitions. You present a number for C2 in Section 5, but that is only applicable to the present data set (i.e. assuming that training accuracy is 1).

Criterion 2 (b) is not clear.
- I neither understand "As the accuracy curve is also monotone decreasing with increasing regularization we will also detect the convexity by a steep drop in accuracy as depicted by the marked point in the Figure 1(b)"
nor do I understand "accuracy over regularization curve (plotted in log-log space) is constant"?
Does that mean that you assume that whenever the training accuracy drops lower than that of the model without regularization, it starts to underfit?

Due to the lack of numerical measures, the experimental evaluation necessarily remains vague by showing some graphs that show that all criteria are roughly met by regularization parameter \lambda=0.00011 on the cifar data set.  In my view, this evaluation of the (vague) criteria is not fit for showing their possible merit.

---

### Official Review · AnonReviewer1 · 2018-11-02
**Not convincing enough.**

**Rating:** 5
**Confidence:** 3

**Review:**

This paper proposed criteria to measure the capacity of a neural network by injecting perturbation (randomized training data). The paper attempted to show that $l_1$-regularization of the kernel weights is a good measure to control the capacity of a network which contradicts the previous finding by Zhang et al (2017) on regularization which claimed that regularization is neither necessary nor by itself sufficient for controlling generalization error.


The proposed method does not require a held out data to check overfitting, which is an interesting direction to explore. The theoretical analysis is seeming to be correct, however, I don’t have strong expertise in theory, therefore, can not assure the correctness.  The experiments, however, are limited. The experiment was done on cifar-10 and the analysis is based on the early stopping, regularization factor and network depth.

There is only one dataset that was used for the experiments, more dataset should be explored for robust evaluation.

The assumptions should be clarified and write clearly. For example, “Thus we also expect that accuracy drops if the regularization of the model is increased.”, which accuracy (training?) and what exactly means by increased regularization (value of $\lambda$)?

---

### Official Review · AnonReviewer2 · 2018-11-03
**Novel contributions are not apparent, needs more empirical evaluation**

**Rating:** 4
**Confidence:** 4

**Review:**

This paper is about detecting overfitting of deep neural networks without using a validation set. This is an interesting research problem. However, it is not clear how this paper contributes to solve the problem. My understanding is that this is a preliminary work, put into a paper in haste. There are more research efforts required to turn it into a good paper, theoretically as well empirically.

One of the key ideas proposed is to obtain multiple instances of neural network models with each one from training on a dataset that is a noisier version of the original dataset; noise is added by permuting lables for a fraction of the original dataset. Then, one can plot training error w.r.t. the level of noise so as to see if the neural model is overfitting.

Authors present their intuitions on what what patterns for the curves (concave curves) would correspond to overfitting. While the arguments seem convincing, one can not be sure unless there is some solid experimental evaluation across multiple datasets or a good theoretical basis.

Here it is also worth noting that the proposed method is not compared w.r.t. any other baseline methods. Basically, in their empirical evaluation, the authors use the existing techniques for regularization to build a variety of neural network models, and then manually analyze the generalization gap for a given model by looking upon the aforementioned curve on training error w.r.t. noise.

Does it mean that the method is just for tuning the values of the parameters related to regularization (like l1 regularization constant, number of iterations, etc)? If so, is there an algorithm to do the fine tuning rather than doing manual analysis of the curves with each one representing a configuration of the regularization parameter values. What would be compute complexity of such an algorithm considering the fact that producing a single curve requires training the neural network multiple times.

---

### Author Response · Authors · 2018-11-27
**Replies to reviewers**

We would like to thank all reviewers for their time and (critical) comments. We appreciate your feed back. Let us answer your concerns.

There does not seem to be a good notion of overfitting for neural network training. This is exemplified for example in the Zhang et al. paper by showing that a network learns data with randomly shuffled labels. Or other memorization experiments, such as Arpit et al. That the authors do comes close to our criterion (C2). The Bartlett et al. and the Liang et al. papers are another example. To show the effects of their (theoretical) methods they analyze the plots of the margin distributions. This is close to (C3).

Our paper tries to give some criteria at hand to show that a proposed method (such as l1-regularization) has an positive effect on the generalization capacity of the trained neural network. In the paper we propose to use criterion (C1) to address this task.

The assumptions made in Section 3.3. serve two purposes: (1) they show that making them results in concave/convex accuracy curves. (2) it connects (C1) to the already used criteria (C2) and (C3).

Looking at the accuracy plots of real curves we see that they are not convex - we attribute this to the dependence of the regularization parameters to the optimization and to a lesser extend correlation within the data.

To make the arguments more convincing we included two further data sets, a different network and analyzed also drop-out and l2-regularization. To summarize the experiments it behaves as claimed, the accuracy plots on mnist looks rather neat.

We further provided a numerical measure for the criteria. We would like to make a point for visual inspection. True the ultimate gaol is an algorithm, but before that we have to be convinced that we are doing something useful. We believe that the margin histograms (C3) and also the plots of (C1) capture overfitting and underfitting quite nicely.

---

### Meta-Review · Area_Chair1 · 2018-12-19

**Confidence:** 5
**Recommendation:** Reject

**Metareview:**

The reviewers reached a consensus that the paper is not fit for publication for the moment because a) the paper lacks thorough experiments and b) the criteria provided by the paper are relatively evague (see more details in reviewer 3's comments.）